# Annotation and Molecular Characterisation of the *TaIRO3* and *TaHRZ* Iron Homeostasis Genes in Bread Wheat (*Triticum aestivum* L.)

**DOI:** 10.3390/genes12050653

**Published:** 2021-04-27

**Authors:** Oscar Carey-Fung, Jesse T. Beasley, Alexander A. T. Johnson

**Affiliations:** School of BioSciences, The University of Melbourne, Melbourne, VIC 3010, Australia; ofung@student.unimelb.edu.au (O.C.-F.); jesse.beasley@unimelb.edu.au (J.T.B.)

**Keywords:** homoeologs, E3-ligase, ubiquitination, hexaploid, KnetMiner, basic helix–loop–helix

## Abstract

Effective maintenance of plant iron (Fe) homoeostasis relies on a network of transcription factors (TFs) that respond to environmental conditions and regulate Fe uptake, translocation, and storage. The iron-related transcription factor 3 (IRO3), as well as haemerythrin motif-containing really interesting new gene (RING) protein and zinc finger protein (HRZ), are major regulators of Fe homeostasis in diploid species like *Arabidopsis* (*Arabidopsis thaliana*) and rice (*Oryza sativa* L.), but remain uncharacterised in hexaploid bread wheat (*Triticum aestivum* L.). In this study, we have identified, annotated, and characterised three *TaIRO3* homoeologs and six *TaHRZ1* and *TaHRZ2* homoeologs in the bread wheat genome. Protein analysis revealed that TaIRO3 and TaHRZ proteins contain functionally conserved domains for DNA-binding, dimerisation, Fe binding, or polyubiquitination, and phylogenetic analysis revealed clustering of TaIRO3 and TaHRZ proteins with other monocot IRO3 and HRZ proteins, respectively. Quantitative reverse-transcription PCR analysis revealed that all *TaIRO3* and *TaHRZ* homoeologs have unique tissue expression profiles and are upregulated in shoot tissues in response to Fe deficiency. After 24 h of Fe deficiency, the expression of *TaHRZ* homoeologs was upregulated, while the expression of *TaIRO3* homoeologs was unchanged, suggesting that TaHRZ functions upstream of TaIRO3 in the wheat Fe homeostasis TF network.

## 1. Introduction

Iron (Fe) is an essential micronutrient for plant growth, as it acts as a protein cofactor in photosynthetic electron transport, chlorophyll biosynthesis, and other cellular processes [1]. Maintaining cellular Fe homeostasis is key to plant health, as toxic concentrations of Fe lead to excess production of reactive oxygen species (e.g., free hydroxyl radicals) via the Fenton reaction [2]. Plants have evolved highly regulated mechanisms that respond to environmental Fe availability and maximise Fe uptake from the soil. Non-graminaceous species primarily use a reduction-based mechanism (traditionally referred to as Strategy I) to convert insoluble ferric Fe (Fe^3+^) to a more soluble ferrous form (Fe^2+^) for direct plant uptake. In contrast, graminaceous species use a chelation-based mechanism (traditionally referred to as Strategy II) involving the secretion of Fe-chelating phytosiderophores (PSs) that bind Fe^3+^ for subsequent PS–Fe^3+^ uptake. Despite these differences in Fe acquisition between non-graminaceous and graminaceous plants, both utilise a conserved, complex network of transcription factors (TFs) that respond to environmental Fe conditions and regulate plant Fe uptake, translocation, and homeostasis [3,4,5,6,7,8].

Master regulators of Fe homeostasis in rice (*Oryza sativa* L.) include the Fe-binding proteins iron deficiency-responsive element-binding Factor 1 (OsIDEF1), haemerythrin motif-containing really interesting new gene (RING), and zinc finger proteins 1 and 2 (OsHRZ1 and OsHRZ2), which sense environmental Fe concentrations and regulate a downstream cascade of TFs [9,10,11]. The OsHRZ1, OsHRZ2, and BRUTUS (the ortholog of HRZ in *Arabidopsis*, AtBTS) proteins are RING-type E3 ligases that polyubiquitinate proteins, such as basic helix–loop–helix (bHLH) TFs, for subsequent degradation via the 26S proteasome [12]. Targets of OsHRZ ubiquitination activity are likely to include the subgroup IVc bHLH TF positive regulators of iron homeostasis 1, 2, and 3 (OsPRI1/OsbHLH60, OsPRI2/OsbHLH58, and OsPRI3/OsbHLH59 respectively), which in turn positively regulate members of subgroups Ib and IVb bHLH TFs [13,14,15]. The subgroup Ib and IVb TFs include the iron-related transcription factor 2 (OsIRO2) and iron-related transcription factor 3 (OsIRO3) in rice, as well as the OsIRO3 ortholog in *Arabidopsis*, POPEYE (AtPYE); these form homodimers or heterodimers capable of binding the promoters of Fe-homeostasis genes. While OsIRO2 proteins positively regulate genes involved in the rice Fe deficiency response, the role of OsIRO3 (and AtPYE in *Arabidopsis*) proteins in the negative regulation of genes involved in Fe uptake and translocation remains poorly understood [16,17,18,19,20].

Bread wheat (*Triticum aestivum* L.) is produced on more land than any other crop, and provides a fifth of all calories consumed by humans [21]. On alkaline soils, which comprise approximately 30% of global soils, Fe is less available for plant uptake and can lead to conditions of Fe deficiency [22]. Under Fe deficiency, wheat plants exhibit leaf chlorosis and reduced yield [23]. Recent progress in bread wheat genetics, such as the release of a fully annotated genome sequence and newly integrated databases, has allowed for efficient in silico gene identification [24,25,26,27,28]. The availability of these genetic resources has recently led to a boom in molecular characterisation efforts of wheat genes involved in Fe transport, chelation, and storage; however, there has been little characterisation of upstream regulators of Fe homeostasis to date [29,30,31,32,33]. Genes from diploid (2n) species, such as rice, often have three orthologous sequences (referred to as homoeologs) in the hexaploid (AABBDD = 6n) bread wheat genome. Characterising all three homoeologs in bread wheat is critical, as approximately 30% of wheat homoeologs show expression bias towards one subgenome [34,35]. Given the complexity of the bHLH TF family proteins (such as IRO2 and IRO3) and the redundancy of the bread wheat genome, the Fe homeostasis network in bread wheat is likely to be more complicated than in related diploid species. Characterising components of the Fe homeostasis network in bread wheat will shed light on this complexity, and may identify novel resources for improving wheat abiotic stress tolerance under Fe-limiting conditions. Here, we have identified, annotated, and characterised the three *TaIRO3*, *TaHRZ1*, and *TaHRZ2* homoeologs in bread wheat, which collectively represent key regulatory genes in the Fe homeostasis network.

## 2. Materials and Methods

### 2.1. Identification and Validation of *TaIRO3* and *TaHRZ* Genes

Identification of the *TaIRO3*, *TaHRZ1*, and *TaHRZ2* genes in bread wheat was performed following the guidelines outline in Bonneau et al. [27]. Briefly, the *OsIRO3* (LOC_Os03g26210), *OsHRZ1* (LOC_Os01g49470), and *OsHRZ2* (LOC_Os05g47780) DNA and protein sequences were used as queries in BLAST searches against the International Wheat Genome Sequencing Consortium (IWGSC) bread wheat reference genome databases (https://plants.ensembl.org/ (accessed on 13 April 2020), https://urgi.versailles.inra.fr/ (accessed on 13 April 2020)), and subsequent putative *TaIRO3* and *TaHRZ* sequences were validated against the rice reference genome database (http://rice.plantbiology.msu.edu/index.shtml (accessed on 13 April 2020)) [36,37,38]. All putative *TaIRO3* and *TaHRZ* gene sequences and associated splice variants were annotated with “high confidence” by the IWGSC (https://plants.ensembl.org/ (accessed on 13 April 2020)) and validated by a bread wheat expressed sequence tags (ESTs) database (https://blast.ncbi.nlm.nih.gov/ (accessed on 17 April 2020)) [25,39]. The *TaIRO3* or *TaHRZ* splice variant that had the highest protein identity to its corresponding homoeologs was considered the main transcript for this study. Naming each *TaIRO3* and *TaHRZ* gene followed the recommended rules for gene symbolization in wheat (http://wheat.pw.usda.gov/ggpages/wgc/98/Intro.htm (accessed on 11 August 2020)) and was based on subgenome and homoeologous grouping.

### 2.2. Phylogenetic Analyses

Wheat, rice, and *Arabidopsis* IRO3 and HRZ protein sequences were used as queries in BLAST searches against the brachypodium (*Brachypodium distachyon*), barley (*Hordeum vulgare* L.), barrel medic (*Medicago truncatula*), common bean (*Phaseous vulgaris* L.), black cottonwood (*Populus trichocarpa*), tomato (*Solanum lycopersicum* L.), potato (*Solanum tuberosum* L.), sorghum (*Sorghum bicolor* L.), red clover (*Trifolium pratense* L.), and maize (*Zea mays* L.) genome databases (https://plants.ensembl.org/) to identify putative IRO3 proteins (labelled bHLH047, IRO3, or bHLH063) and HRZ proteins (labelled BTS or HRZ) in a range of eudicot and monocot species [38]. The putative barley HvHRZ2 protein sequence was identified through BLAST searches against the barley reference genome database (https://webblast.ipk-gatersleben.de/barley_ibsc/ (accessed on 19 August 2020)) [40,41]. The putative IRO3 and HRZ proteins were validated against the wheat (https://plants.ensembl.org/ (accessed on 19 August 2020)), rice (http://rice.plantbiology.msu.edu/index.shtml (accessed on 19 August 2020)), and *Arabidopsis* (https://www.arabidopsis.org/ (accessed on 19 August 2020)) reference genome databases prior to phylogenetic analysis [36,42]. Sequence alignment of eudicot and monocot IRO3 and HRZ proteins was performed using Geneious software (https://www.geneious.com/v11.0.6 (accessed on 20 August 2020)), and phylogenetic construction was performed using the PhyML Geneious plugin with the LG substitution model and a bootstrap value of 1000 [43]. Conserved protein domains within IRO3 and HRZ proteins were predicted using InterPro EMBL-EBI software (https://www.ebi.ac.uk/interpro/ (accessed on 27 August 2020)) [44]. The IRO3 and HRZ proteins used in sequence alignment and phylogenetic analysis are provided in Appendix A.

### 2.3. Bread Wheat Tissue Sampling and Quantitative Reverse Transcription-PCR (qRT-PCR) Analyses of the *TaIRO3* and *TaHRZ* Genes

To assess tissue-specific expression of the *TaIRO3* and *TaHRZ* genes in the bread wheat cultivar (cv.) Chinese Spring, a catalogue of cDNA from ten different tissues at various developmental stages was generated, as described by Schreiber et al. [45]. Briefly, total RNA was extracted from tissues harvested from 7–10 plants to produce three independent biological samples of cDNA. The tissues analysed in this study included embryonic root and mesocotyl (two-day-old embryos); seedling root, crown, and seedling leaves collected 10–12 days after sowing; bracts, anthers, and pistils (prior to anthesis); and caryopsis and embryos collected 3–5 and 22 days after pollination (DAP), respectively.

To determine how *TaIRO3* and *TaHRZ* gene expression is influenced by environmental Fe conditions, total RNA was extracted and cDNA generated from bread wheat cv. Gladius shoot and root tissues, as described in Bonneau et al. [30]. Briefly, wheat seedlings were grown in 20 L tubs containing a nutrient-replete, hydroponic growth solution for three weeks before transfer to an Fe deficiency treatment for one week. During the Fe deficiency treatment period, wheat plants were subjected to either Fe sufficient (50 µM) or Fe deficient (0 µM) conditions, with shoot and root tissues harvested from three individual wheat plants at days 0, 1, 5, and 7 of the treatment period.

Quantitative reverse-transcription PCR (qRT-PCR) analyses of the three *TaIRO3*, *TaHRZ1*, and *TaHRZ2* genes in the tissues of bread wheat cultivars Chinese Spring and Gladius were carried out using subgenome-specific primers (Appendix A) that were validated using Chinese Spring nulli-tetrasomic DNA [46]. Relative expression for each *TaIRO3* and *TaHRZ* gene was calculated based on a standard curve of triplicate, 10-fold serial dilutions (10^1^–10^7^) of purified template for each primer pair, and were based on the amount of RNA (µg) included in the reaction. A three-gene normalisation factor (3GNF) of the wheat house-keeping genes—*cyclophilin* (*TaCyc*), *actin* (*TaActin*), and *elongation factor 1-α* (*TaEFA*) for cv. Chinese Spring, and *TaCyc*, *TaActin*, and *glyceraldehyde 3-phosphate dehydrogenase* (*TaGAPDH*) for cv. Gladius—was used to normalise *TaIRO3* and *TaHRZ* qRT-PCR gene expression data [47]. Expression data for *TaIRO3* and *TaHRZ* genes was normalised independently for shoot and root tissues in cv. Gladius. The relative expression level of each *TaIRO3*, *TaHRZ1*, and *TaHRZ2* homoeolog was compared between subgenomes, using the ternary plot function in the Wheat Expression Browser database (wheat-expression.com (accessed on 1 April 2021)) [35,48] (Appendix A).

### 2.4. Gene Network Construction

A *TaIRO3*, *TaHRZ1* and *TaHRZ2* gene network was generated using the bread wheat database of KnetMiner software (https://knetminer.org/β/Triticum_aestivum/ (accessed on 4 September 2020)), which compiles a network of bread wheat genes and related orthologs from *Arabidopsis* and rice. Regulatory associations between bread wheat transcription factors and our queries were predicted by Genie3 software, which was recently integrated into KnetMiner [49]. A complete network containing all the phenotypic traits and molecular functions associated with the *TaIRO3* homoeologs and *TaHRZ* gene family is available at https://knetminer.com/β/knetspace/network/f6c84de2-92ae-45e3-a068-0e61419b24cb, and all bread wheat gene identification numbers are provided in Appendix A.

### 2.5. Statistical Analysis

Statistically significant differences in cv. Gladius tissue gene expression under Fe-sufficient or Fe-deficient conditions was determined at each time point of the treatment period using a two-sample Student’s *t*-test (assuming equal variances) in Minitab software (http://www.minitab.com/en-us/v19.0 (accessed on 28 April 2020)). Graphs were generated using the ggplot2 package in RStudio (https://rstudio.com/v3.6.3 (accessed on 28 April 2020)).

## 3. Results

### 3.1. Three *TaIRO3* Genes Are Located on Chromosomal Group 2, and Six *TaHRZ* Genes Are Located on Chromosomal Groups 1 and 3

Three *TaIRO3* homoeologous genes were identified on chromosomal group 2 of the bread wheat genome, with one *TaIRO3* homoeolog located on each of the A, B, and D subgenomes (hereafter referred to as *TaIRO3-A*, *TaIRO3-B*, and *TaIRO3-D*, respectively). The *TaIRO3* genomic sequences ranged between 2379 to 2679 base pairs (bp), due to differences in intron length (Figure 1). Gene structure was conserved across the *TaIRO3* homoeologs, except for *TaIRO3-B*, which possesses a truncated coding region and an intron within its 5′UTR. Genomic sequence identity within the *TaIRO3* homoeologs was between 76.9% to 79.8%. Three *TaHRZ1* genes were identified on chromosomal group 3, and three *TaHRZ2* genes were identified on chromosomal group 1 of the bread wheat genome. One *TaHRZ1* and one *TaHRZ2* homoeolog were identified on each of the A, B, and D sub-genomes (hereafter referred to as *TaHRZ1-A*, *TaHRZ1-B*, *TaHRZ1-D*, *TaHRZ2-A*, *TaHRZ2-B*, and *TaHRZ2-D*, respectively). The *TaHRZ1* genomic sequences ranged between 11,406 to 12,345 bp, and the *TaHRZ2* genomic sequences ranged between 9116 to 9585 bp, due to differences in intron length (Figure 2). Gene structure was conserved among the *TaHRZ* gene family, with genomic sequence identity ranging between 88.9% to 95.9% for the *TaHRZ1* homoeologs and 83.3% to 93.1% for the *TaHRZ2* homoeologs. In addition to the six *TaHRZ* genes, we identified 36 predicted splice variants within the *TaHRZ* gene family (Appendix A).

### 3.2. IRO3 Proteins Are Conserved within Graminoids and Separate between Monocots and Eudicots

Phylogenetic analysis of IRO3 proteins from a range of monocots (referred to as IRO3 or bHLH063) and eudicots (referred to as PYE or bHLH047) identified two groups of IRO3 proteins (Figure 3). The TaIRO3 proteins were most closely related to the barley IRO3 (HvIRO3) protein, ranging between 271 to 339 amino acids (aa) in length and shared between 74.5% to 90.8% identity (Figure 3a,b). All IRO3 proteins contained the highly conserved HLH domain of bHLH-type TFs, which functions in protein dimerisation and DNA binding (Appendix A). A conserved, ethylene-responsive, element binding factor-associated amphiphilic repression (EAR) or “EAR-like” motif was identified in both eudicot (DLNxxP or VLNxxP) and monocot (LxLxL or RxLxL) IRO3 proteins.

### 3.3. HRZ Proteins Are Highly Conserved and Separate into Two Clades in Graminaceous Species

Phylogenetic analysis of HRZ proteins from a range of monocots and eudicots (referred to as BTS) identified two groups of HRZ proteins (Figure 4). The graminaceous species within monocots further separates into two clades, with the TaHRZ1 proteins forming one clade with HvHRZ1, BdHRZ1, OsHRZ1, SbHRZ1, and ZmHRZ1, and the TaHRZ2 proteins forming a second clade with HvHRZ2, BdHRZ2, OsHRZ2, SbHRZ2, and ZmHRZ2 (Figure 4a). The TaHRZ1 proteins were all 1237 aa in length and shared between 97.8% to 99.1% identity, and the TaHRZ2 proteins ranged between 1239 to 1242 aa in length and shared between 95.5% to 98.1% identity (Figure 4b). All HRZ proteins contained at least one hemerythrin domain, with the TaHRZ proteins containing three hemerythrin domains each. All HRZ proteins contained a conserved zinc (Zn)-finger region (comprised of CHY-type, CTCHY-type, and RING-type motifs), and all HRZ proteins (except for SbHRZ1) contained a RCHY1 Zn-ribbon domain near the C-terminus (Appendix A).

### 3.4. The *TaIRO3*, *TaHRZ1*, and *TaHRZ2* Homoeologs Have Distinct Expression Patterns in Bread Wheat Tissues and Are Upregulated in Response to Fe Deficiency

Gene expression analysis of the *TaIRO3*, *TaHRZ1,* and *TaHRZ2* genes across a range of bread wheat cv. Chinese Spring tissues and developmental stages revealed distinct expression patterns (Figure 5). The *TaIRO3* genes were broadly expressed across a range of tissues, with the *TaIRO3-A* homoeolog being expressed higher in the anthers (2.6- to 3.0-fold), caryopsis (23- to 24-fold), and embryo (10- to 13-fold) relative to the *TaIRO3-B* and *TaIRO3-D* homoeologs (Figure 5a). The *TaHRZ* gene family was highly expressed in leaves, bracts, and anthers relative to other tissues, with the expression of *TaHRZ1-A* between 1.5- and 4.0-fold higher than *TaHRZ1-B* and *TaHRZ1-D*, and the expression of *TaHRZ2-D* between 1.4- and 4.8-fold higher than *TaHRZ2-A* and *TaHRZ2-B* across all tissues (Figure 5b,c). Similar patterns of relative expression between *TaIRO3*, *TaHRZ1*, and *TaHRZ2* homoeologs was observed in the Wheat Expression Browser database (Appendix A). Expression of the *TaIRO3-A* homoeolog was significantly upregulated in bread wheat cv. Gladius shoot tissues at day 1, 5, and 7, and expression of the *TaIRO3-B*, and *TaIRO3-D* homoeologs was significantly upregulated in shoot tissues at day 5 and 7 of the Fe deficiency treatment (Figure 6). Within the Fe deficiency treatment, expression of the *TaIRO3* genes in shoot tissues was upregulated (between 2.0- and 2.9-fold) from day 1 to day 7. In root tissues, the expression of *TaIRO3-B* was significantly upregulated at day 5, and expression of *TaIRO3-A* and *TaIRO3-D* was significantly upregulated at day 7 of the Fe deficiency treatment (Figure 6). Expression of the *TaHRZ* genes (except for *TaHRZ1-D*) was significantly upregulated in shoot tissues at day 1, and expression of the *TaHRZ2-B* homoeolog was significantly upregulated in shoot tissues at day 7 of the Fe deficiency treatment (Figure 6). Within the Fe deficiency treatment, expression the *TaHRZ* genes was downregulated (between 1.5- to 3.2-fold) from day 1 to day 7. In root tissues, expression of *TaHRZ1-A*, *TaHRZ2-A*, and *TaHRZ2-D* was significantly upregulated at day five of the Fe deficiency treatment (Figure 6).

### 3.5. The *TaIRO3* and *TaHRZ* Genes Are Associated with Regulatory Components of Fe Homeostasis in Arabidopsis and Rice

KnetMiner analysis of the *TaIRO3* and *TaHRZ* gene sequences confirmed the presence of three *TaIRO3*, three *TaHRZ1*, and three *TaHRZ2* homoeologs in the bread wheat genome. A KnetMiner network revealed regulatory associations between predicted TaIRO3 proteins and OsbHLH062 in rice (a subgroup IVb TF), as well as regulatory associations between predicted TaHRZ proteins and AtBTS (Figure 7). The predicted TaHRZ proteins were also associated with the rice stress-related ring finger protein 1 (OsSRFP1). The network showed physical interactions between AtBTS and the subgroup IVc TFs AtbHLH115, AtILR3 and AtbHLH104. Genie3 software uncovered putative TFs belonging to subgroup Ib, ABI3/VP1 (TaABI3), and other putative TFs (TaARF1) predicted to have regulatory associations with *TaHRZ* and *TaIRO3* genes. 

## 4. Discussion

Understanding how Fe is regulated in wheat and other staple crops is essential for maximising crop production under Fe-limiting, alkaline soil conditions. Identifying regulatory components of Fe homeostasis in wheat was difficult until the recent release of a fully annotated bread wheat genome and integrated databases [24,25,26]. Here, we identified, annotated, and characterised the *TaIRO3*, *TaHRZ1* and *TaHRZ2* genes in bread wheat, which together act to regulate Fe homeostasis transcriptionally and post-transcriptionally. The TaIRO3 proteins belong to the subgroup IVb bHLH TF family, one of 26 subgroups that are composed of more than 147 bHLH TFs in plants, and regulate transcription by dimerising and binding E-boxes (CANNTG) in gene promoters via a basic region of the HLH domain (Figure 3b) [50,51,52]. Once bound to DNA, IRO3 proteins are thought to repress transcription by recruiting a histone deacetylating complex via an ethylene-responsive, element binding factor-associated, amphiphilic repression (EAR) motif [53]. The EAR motif commonly has consensus sequence DLNxxP or LxLxL, and here we observed conservation of the DLNxxP motif at amino acid position 221 in eudicots (Appendix A). In contrast, the EAR motif in monocot IRO3 proteins was conserved at amino acid position 286, and had the sequence LxLxL (in ZmIRO3, SbbHLH063 and OsIRO3 proteins) or RxLxL (all other monocots, including the TaIRO3 proteins). A subgroup IVb bHLH TF in *Arabidopsis*, AtbHLH11, was also recently described to have a functional LxLxL EAR motif near amino acid position 286 [54]. Given that RxLxL is an atypical EAR motif sequence, we have labelled this sequence as “EAR-like” in the TaIRO3 proteins, and further research is required to confirm whether this motif is functional in bread wheat. In contrast to the TaIRO3 proteins, the TaHRZ proteins belong to an E3-ligase family that contain conserved, putative metal-binding domains (hemerythrin, CHY-type, CTCHY-type, RING-type, and RCHY1 Zn-ribbon) that together act to post-transcriptionally regulate genes [10,55]. All HRZ proteins contained a conserved Zn-finger region (containing CHY-type, CTCHY-type, and RING-type motifs) at the C-terminus that is responsible for polyubiquitination and the subsequent degradation of proteins (Figure 4b). In addition to the Zn-finger region, all HRZ proteins (except for SbHRZ1) contain a conserved RCHY1 Zn-ribbon domain (also known as a rubredoxin-type fold), and investigating whether there is variation in post-transcriptional regulation between *S. bicolor* (which lacks the RCHY1 Zn-ribbon domain) and other monocots is warranted (Appendix A). All HRZ proteins also contain at least one hemerythrin domain, with TaHRZ proteins containing three hemerythrin domains, which bind to Fe, Zn, and/or oxygen atoms, resulting in changes to protein stability [11,56]. The eudicot BTS-like (BTSL) family of proteins contain only two hemerythrin domains and perform a separate role to BTS in Fe homeostasis, suggesting that the hemerythrin domains may be important in the regulation of Fe homeostasis, although this requires further investigation [57]. We identified several splice variants of the *TaHRZ1* genes that encode proteins with a reduced number of hemerythrin domains or missing motifs within the Zn-finger region (Appendix A). As alternative gene splicing plays a key role in response to Fe stress in *Arabidopsis* and rice, these *TaHRZ* splice variants may be critical to maintaining Fe homeostasis in wheat [58,59,60]. Furthermore, the truncated TaHRZ proteins encoded by these splice variants provide a novel resource for confirming the role of hemerythrin domains and the Zn-finger region within the TaHRZ proteins.

The *TaIRO3*, *TaHRZ1*, and *TaHRZ2* genes share patterns of gene expression in bread wheat tissues that suggest they are part of the same Fe homeostasis regulatory network. We detected high expression of the *TaHRZ* genes and the *TaIRO3-A* homoeolog in wheat anther tissues (Figure 5), which contain high levels of Fe and Zn that are essential for pollen germination [61]. Genes involved in the biosynthesis of the Fe chelators nicotianamine (NA) and 2′deoxymugineic acid (DMA) are also highly expressed in bread wheat anthers, suggesting that the high concentrations of anther Fe requires tight regulation in order to avoid oxidative stress and tissue damage [29,30]. Interestingly, high expression of the *TaIRO3-A* homoeolog (relative to *TaIRO3-B* and *TaIRO3-D*) in the developing wheat embryo (Figure 5a) suggests that homoeologous wheat genes may perform tissue-specific functions, and similarly, high levels of *TaIRO3* expression in the wheat grain were observed in a USDA wheat expression database (https://wheat.pw.usda.gov/WheatExp/ (accessed on 28 April 2020)) [62,63]. The *TaHRZ* gene family was highly expressed in shoot tissues throughout the wheat lifecycle (Figure 5 and Figure 6), suggesting that TaHRZ proteins regulate Fe homeostasis more in wheat leaf tissues relative to wheat root tissues; a similar shoot-specific function for OsHRZ proteins has been hypothesised in rice [10]. The OsHRZ proteins in rice root tissues are degraded, and may therefore result in the upregulated expression of genes related to Fe uptake and translocation (i.e., NA synthase, NA aminotransferase, etc.) in rice roots. Under conditions of Fe deficiency, all *TaIRO3* and *TaHRZ* genes were upregulated in shoot tissues (Figure 6), although at different timepoints. All members of the *TaHRZ* gene family (except for *TaHRZ1-D*) were upregulated in wheat shoot tissues within 24 h of exposure to the Fe deficiency treatment, whereas upregulation of the *TaIRO3* genes in shoot tissues did not occur until day 5 of Fe deficiency (Figure 6). Together, these results provide evidence that *TaHRZ* acts upstream of *TaIRO3* in the wheat Fe deficiency response, which is supported by gene expression and biochemical evidence that *OsHRZ* regulates *OsIRO3* in the rice Fe deficiency response [10,13,18]. The expression of all *TaHRZ* genes in wheat shoot tissues was downregulated after 24 h of Fe deficiency, whereas the expression of *OsHRZ* in rice shoot tissues is upregulated up to day 7 of Fe deficiency [10]. Together, these results point to key differences in the regulation of Fe homeostasis between rice and wheat, and suggest that *TaHRZ* genes may form a negative feedback loop in wheat shoot tissues. We propose the TaHRZ proteins in wheat are pivotal for managing Fe homeostasis, and that downregulation of the *TaHRZ* genes after 24 h of Fe deficiency allows for upregulation of Fe deficiency response genes (including *TaIRO3* and NA/DMA biosynthesis genes), to ensure wheat plants can absorb sufficient Fe for growth [29,30,31]. Our KnetMiner/Genie 3 network confirms that all *TaHRZ* and *TaIRO3* genes are distinct homoeologs present in the bread wheat genome and identifies regulatory associations predicted from gene expression data between the *TaHRZ* and *TaIRO3* genes, with five genes belonging to the subgroup Ib bHLH TF family (including a putative *OsIRO2* ortholog) and three genes belonging to the ABI3/VP1 TF family in bread wheat (Figure 7) [35,49]. These *ABI3/VP1* genes could be related to the master Fe sensor *OsIDEF1* in rice, and future analysis will aim to characterise these novel TFs in bread wheat and determine whether the *TaABI3/VP1* genes share similar Fe sensing roles to *OsIDEF1* [64]. The KnetMiner network includes a different rice bHLH protein to OsIRO3, OsbHLH062, likely due to incomplete incorporation of the rice genome database into KnetMiner. We anticipate that the *OsbHLH062* gene encodes a closely related TF to OsIRO3 in rice, and therefore warrants further investigation in future rice Fe homeostasis studies. The Genie3 software did not identify any subgroup IVc bHLH TFs in our KnetMiner network, likely due to Genie3 using RNA-seq datasets, and that IVc bHLH TFs (including OsPRI1/2/3) are not regulated at the transcriptional level [13,14,15].

Loss- and gain-of-function studies in rice and *Arabidopsis* have demonstrated that HRZ and IRO3 proteins repress the Fe deficiency response, and suggest that similar approaches coupled with *in planta* protein analyses are now required to confirm TaHRZ and TaIRO3 function in bread wheat [10,17,18,19,20]. No knockdowns of *OsIRO3* or *AtPYE* have been reported to date; however, a knockout of *OsIRO3* in rice upregulated the expression of Fe deficiency response genes in rice roots under Fe sufficiency, and resulted in Fe toxicity in shoot tissues under Fe deficiency [19,20]. Knockout of *AtBTS* in *Arabidopsis* is embryonically lethal; however, knockdowns of *AtBTS* and *OsHRZ* in *Arabidopsis* and rice, respectively, have enhanced Fe deficiency tolerance and increased grain Fe concentration [10,65]. Knocking out individual *TaIRO3*/*TaHRZ* homoeologs within the bread wheat genome may be less detrimental than in diploid species, due to genetic redundancy in the hexaploid genome, and may instead mimic the effect of knocking down *OsIRO3/AtPYE* or *OsHRZ/AtBTS* genes in rice and *Arabidopsis*. The *TaIRO3* and *TaHRZ* genes identified in this study contribute to our understanding of Fe homeostasis in bread wheat, and provide a novel resource for marker-assisted selection, genome editing, and genetic modification of bread wheat to improve abiotic stress tolerance and increase grain Fe concentration.

## Figures and Tables

**Figure 1 genes-12-00653-f001:**
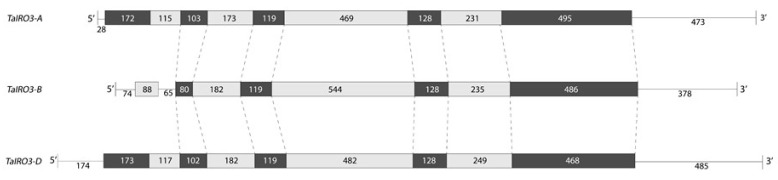
Gene structure of the *TaIRO3* homoeologs in bread wheat cv. Chinese Spring. Coding sequences (black boxes), introns (grey boxes), and known untranslated regions (lines) of *TaIRO3* homoeologs are depicted with length (base pairs) provided.

**Figure 2 genes-12-00653-f002:**
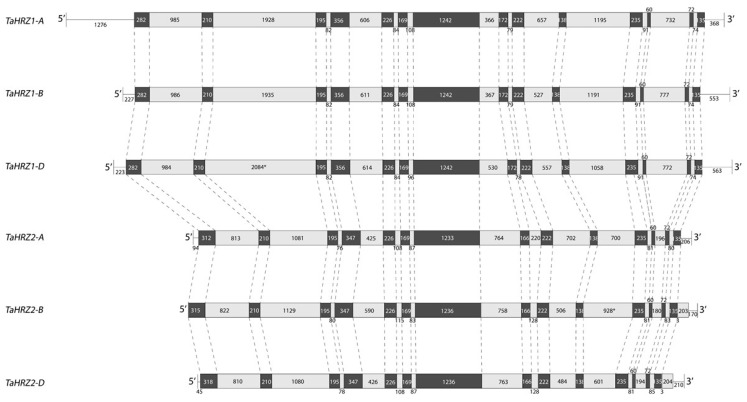
Gene structure of the *TaHRZ1* and *TaHRZ2* homoeologs in bread wheat cv. Chinese Spring. Coding sequences (black boxes), introns (grey boxes), and known untranslated regions (lines) of *TaHRZ1* and *TaHRZ2* homoeologs are depicted with lengths (base pairs) provided. Asterisk indicates introns of estimated length.

**Figure 3 genes-12-00653-f003:**
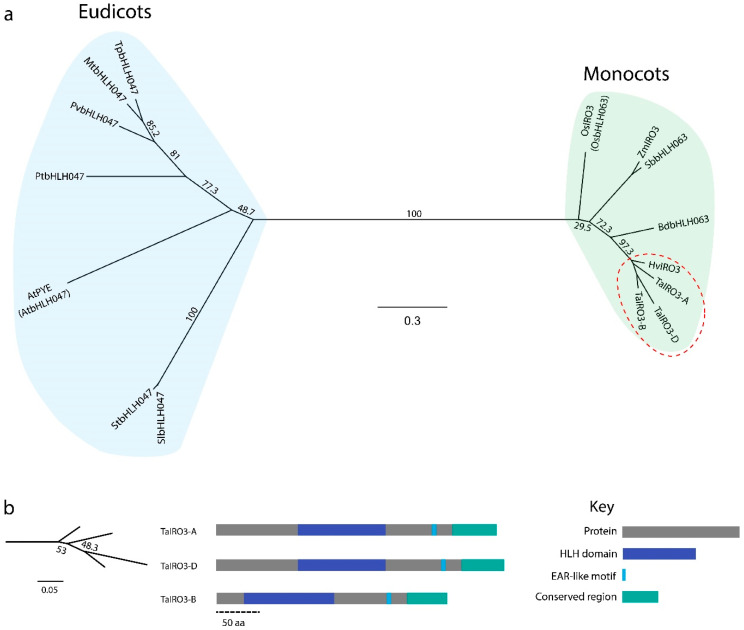
Phylogenetic comparison of IRO3 proteins in higher plants. (**a**) An unrooted phylogenetic tree of IRO3/PYE proteins, showing eudicot and monocot groups (light blue), as well as a graminiod subgroup (green). The red circle indicates the IRO3 proteins in Triticeae. (**b**) The structure of TaIRO3 proteins with conserved domains highlighted. The region in green labelled “Conserved region” has an unknown function. Dashed scale bar indicates length in amino acids. Numbers indicate bootstrap percentage, and scale bars represent evolutionary distance in substitutions per site. Abbreviations: *Arabidopsis thaliana* (At), *Brachypodium distachyon* (Bd), *Hordeum vulgare* L. (Hv), *Medicago truncatula* (Mt), *Oryza sativa* L. (Os), *Phaseolus vulgaris* L. (Pv), *Populus trichocarpa* (Pt), *Solanum lycopersicum* L. (Sl), *Solanum tuberosum* L. (St), *Sorghum bicolor* L. (Sb), *Trifolium pratense* L. (Tp), *Triticum aestivum* L. (Ta), and *Zea mays* L. (Zm).

**Figure 4 genes-12-00653-f004:**
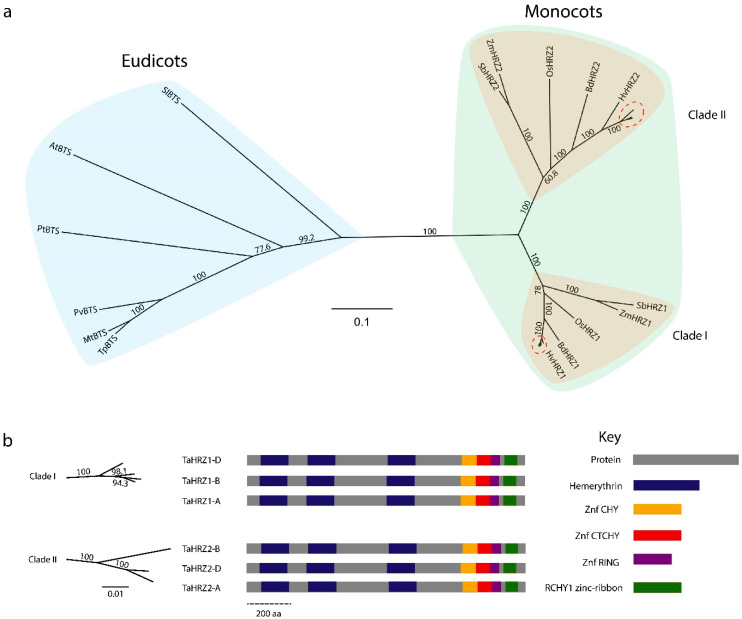
Phylogenetic comparison of HRZ1 and HRZ2 proteins in higher plants. (**a**) An unrooted phylogenetic tree of HRZ1 and HRZ2 proteins, showing eudicot (light blue) and graminiod (green) groups, and the two clades within graminoids (light brown). Red circles indicate the HRZ1 proteins in *Triticeae* and the TaHRZ2 proteins in wheat. (**b**) The structure of TaHRZ1 and TaHRZ2 proteins with conserved domains is highlighted. Dashed scale bar indicates length in amino acids. Numbers indicate wheat bootstrap percentage, and scale bars represent evolutionary distance in substitutions per site. Abbreviation: *A. thaliana* (At), *B. distachyon* (Bd), *H. vulgare* L. (Hv), *M. truncatula* (Mt), *O. sativa* L. (Os), *P. vulgaris* L. (Pv), *P. trichocarpa* (Pt), *S. lycopersicum* L. (Sl), *S. bicolor* L. (Sb), *T. pratense* L. (Tp), *T. aestivum* L. (Ta), and *Z. mays* L. (Zm).

**Figure 5 genes-12-00653-f005:**
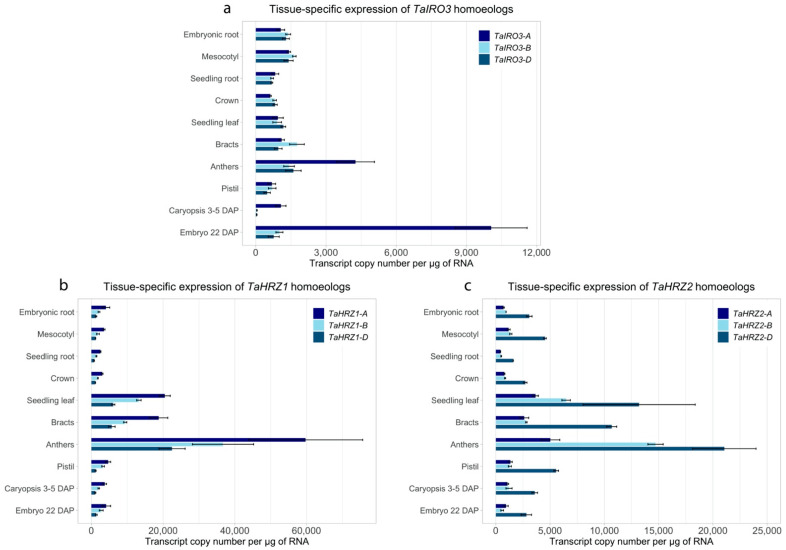
Quantitative reverse-transcription PCR analysis of *TaIRO3*, *TaHRZ1*, and *TaHRZ2* homoeologs within 10 different tissues and developmental stages of bread wheat cv. Chinese Spring. The relative expression (transcript copy number per µg of RNA) is provided for each homoeolog of (**a**) *TaIRO3*, (**b**) *TaHRZ1*, and (**c**) *TaHRZ2*. Error bars indicate standard deviation of the mean of three technical replicates derived from a bulk of 7–10 independent samples.

**Figure 6 genes-12-00653-f006:**
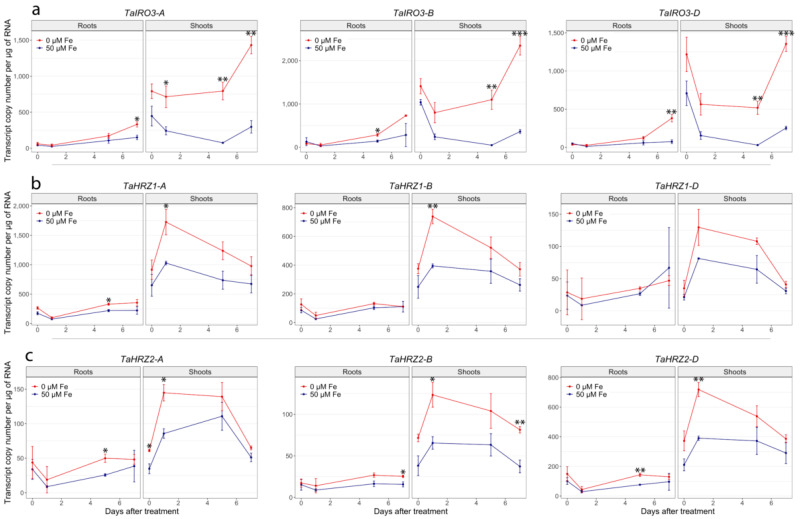
Quantitative reverse-transcription PCR analysis of *TaIRO3*, *TaHRZ1,* and *TaHRZ2* homoeologs within shoot and root tissues of bread wheat cv. Gladius under Fe sufficient or deficient conditions. The relative expression (transcript copy number per µg of RNA) is provided for each homoeolog of (**a**) *TaIRO3*, (**b**) *TaHRZ1*, and (**c**) *TaHRZ2* in root and shoot tissues. Timepoints correspond to 0 (experiment start), 1, 5, or 7 days of growth under Fe sufficient (+Fe, blue line) or Fe deficient (−Fe, red line) conditions. Error bars indicate the SEM of three biological replicates of each homoeolog (*n* = 3). Asterisks indicate significant differences between Fe deficient and sufficient conditions at each time point (two-sample Student’s *t*-test assuming equal variance; * = *p* value ≤ 0.05; ** = *p* value ≤ 0.01; *** = *p* value ≤ 0.001).

**Figure 7 genes-12-00653-f007:**
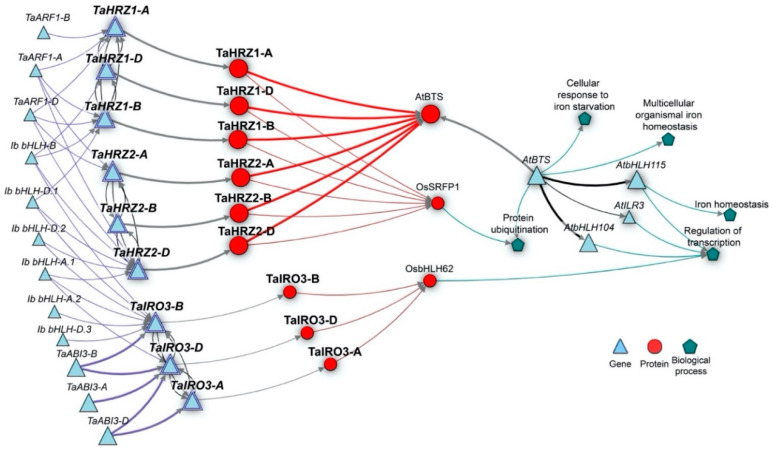
Condensed Knetminer gene network of *TaIRO3*, *TaHRZ1* and *TaHRZ2* homoeologs. Novel transcription factor associations are predicted with Genie3. Blue lines represent regulatory associations. Thick blue lines represent associations between ABI3/VP1 transcription factor family and the *TaIRO3* homoeologs. Transcription factors with fewer than two links were excluded. Grey lines represent a gene encoding a protein. Red lines indicate a similar protein sequence. Teal lines represent a role in a biological process. Black lines indicate physical interactions. Lines between *TaHRZ* and *TaIRO3* gene families represents homoeologous relationship.

## Data Availability

The data presented in this study are openly available in FigShare at https://doi.org/10.26188/14483346.v1.

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
