# Peer review of "Annotation and Molecular Characterisation of the TaIRO3 and TaHRZ Iron Homeostasis Genes in Bread Wheat (Triticum aestivum L.)"

_genes, 2021, doi:10.3390/genes12050653_

Round 1

Reviewer 1 Report

The manuscript by Carey-Fung and colleagues describes the wheat gene copies of an iron-regulated transcription factor and of a subfamily of E3 ubiquitin ligases. The information is presented very clearly, and the qPCR expression data are of high quality. However, more in-depth analysis is needed to make it useful for the community and justify the manuscript as a standalone publication.

  1. Please reword the title to make it clear that this paper does not physiologically characterise the roles of TaIRO3 and TaHRZ in iron homoeostasis. I would suggest something more along the lines of “Annotation and expression analysis of the TaIRO3…”. Similarly, in the abstract it is premature to say that the genes have been “characterised and validated” (line 14-15).
  2. The aim / key question of the paper should be more precisely worded in the Abstract, which then allows you to better formulate a concluding sentence at the end of the Abstract. The text “The Fe homeostasis TF network is well characterised in [..] Arabidopsis and rice but remains poorly understood in hexaploid bread wheat” is rather broad and could be the basis for many research papers, or even reviews. Am I correctly assuming that IRO3 and HRZ are of interest as potential candidate genes for biofortification approaches?
  3. Following on from the previous point, it is of interest to note that the core regulatory mechanism of Fe homeostasis appears to be conserved between monocots and dicots. For many decades, plants were strictly separated into Strategy I and II, because the iron uptake mechanisms were so different. Several recent review papers by the groups of Dubos and Guerinot, see special issue of J Exp Bot that came online in the past couple of months, describe the common regulatory actors, which include IRO3 and HRZ/BTS. There may be other recent papers as well, so please use these to update refs 3 – 6 and relevant sections in the Introduction.
  4. Line 182: It is inappropriate to claim that there is a separation between graminaceous and non-graminaceous species when the only example of the latter is banana. I would suggest you incorporate other non-graminaceous monocot species in the phylogenetic analysis, which would then provide you with the power to make this distinction (or alternatively remove this distinction from your results).
  5. Further to point 5, it would be worth adding the BTS-like proteins in dicots to the HRZ/BTS phylogenetic tree.
  6. What exactly does “transcript copy number per ug of RNA” refer to in Fig 5 and 6? I have not seen this scale used for qRT-PCR previously, and it is not clear to me how this expression data incorporates normalisation by the three housekeeping genes in the methods. Can you please make this calculation explicit in the methods, and explain the scale used in the figure captions.
  7. You make the claim that HRZ is upregulated at 24 hours in shoot tissue after the initial iron deprivation (Figure 6), but it also looks to me that the HRZ genes are upregulated in the control environment as well, though perhaps to a lesser extent. Have you carried out comparisons across the timepoints to see whether the expression of the IRO3 and HRZ genes varies across the timecourse for both conditions? From looking at your data, it seems as if the HRZ genes in particular are upregulated 24 hours after the start of the experiment in both Fe deficient and Fe sufficient conditions, suggesting that the response may also in part be due to the transplanting/moving of the plants.
  8. The section looking at the KnetMiner network needs to be fleshed out more. What kinds of “associations” do you see (lines 265-267)? Looking at the network online, these seem to be based on sequence similarity so this part of the network doesn’t provide any new information beyond your phylogenetic analysis for the HRZ genes. I am also surprised that the KnetMiner network pulls out a different rice bHLH gene as most similar to the IRO3 genes. Have you looked into this bHLH062 gene any further? Is this gene equally orthologous to the wheat IRO3 genes? It would be good to know if this gene appeared in your initial BLAST identification of the IRO3 genes, and how you were able to determine that bHLH063 (OsIRO3) is the true ortholog of the TaIRO3 genes, rather than bHLH062. It’s important to note that the majority of the data you report from the KnetMiner network is not novel (either known from your phylogeny, or is reporting knowledge from the Arabidopsis literature). The most interesting aspect here is looking at the predicted regulation of the HRZ and IRO3 gene by the GENIE3 network. I would suggest you emphasize this aspect of the analysis much more both in the results and in the discussion.
  9. Given your interest in the global expression of the IRO3 and HRZ genes, as well as the genome dominance they may show, I would suggest you look at the extensive expression data available on wheat-expression.com. Not only can you see expression data for the 9 genes in question, but you can visualise the homoeolog dominance using the “ternary plot” option. I’ve had a quick look at the data for your genes and it confirms your results, and would give strength to your claims about the expression profile of the IRO3 and HRZ genes.
  10. Lines 356-358: I am not sure how the GENIE3 network using RNA-Seq data is relevant to no subgroup IVc bHLH transcription factors being identified in the KnetMiner network. Also, please be aware that the GENIE3 network does not contain any iron-responsive datasets, so this would have no bearing on their inclusion in the network.

Minor Points:

  1. Introduction, line 59: Wheat grows extremely well on alkaline soils and Fe uptake is never an issue, to my knowledge. The only agronomy sectors where Fe fertilization is consistently applied is for fruit trees (peach, apricot, cherry). Please rephrase this section.
  2. Please reference the primer supplementary table in the methods section.
  3. The genomes and resources used in the methods section should be cited, rather than just referenced with a URL. For example, in lines 81 and 82, include the appropriate citations for the wheat genome (Appels et al. 2018) and EnsemblPlants (Howe et al. 2019). Please include appropriate citations for the remaining tools and genomes used in the methods section.
  4. The TaIRO3-B homeologue seems to be missing the first exon. Is this really correct, or due to incomplete genome sequence?
  5. Panel C in Figures 3 and 4 is very small and difficult to read. I would suggest moving panel C from both figures into a joint supplementary figure, which allows the alignment to be larger. I would also suggest providing the given alignment as a supplementary file, which would support reconstruction of the phylogenetic analyses.
  6. The Discussion section is rather long given the small data set. Consider merging Results and Discussion, and restrict “Discussion” to placing the data in context, e.g. comparing the expression with exVIP data, and with recent RNA-seq data of wheat grown -Fe.

Author Response

Reviewer 1 Statement:

The manuscript by Carey-Fung and colleagues describes the wheat gene copies of an iron-regulated transcription factor and of a subfamily of E3 ubiquitin ligases. The information is presented very clearly, and the qPCR expression data are of high quality. However, more in-depth analysis is needed to make it useful for the community and justify the manuscript as a standalone publication.

Reviewer 1 comments:

Comment 1:

Please reword the title to make it clear that this paper does not physiologically characterise the roles of TaIRO3 and TaHRZ in iron homoeostasis. I would suggest something more along the lines of “Annotation and expression analysis of the TaIRO3…”. Similarly, in the abstract it is premature to say that the genes have been “characterised and validated” (line 14-15).

We agree that the title is ambiguous and does not completely reflect the nature of our manuscript. To clarify, we have tweaked the title to read ‘Annotation and molecular characterisation of the TaIRO3 and TaHRZ iron homeostasis genes in bread wheat (Triticum aestivum L.)’ (Page 1, Lines 2-4), adding the word ‘annotation’, clarifying the type of characterisation we performed and removing the word ‘regulating’. We believe this new title reflects the true nature of our manuscript which characterized (in terms of annotation, phylogeny, protein analysis, and tissue-specific expression) the well-known iron homeostasis
genes: IRO3 and HRZ in bread wheat. We have also added the word ‘annotated’ and removed the word ‘validated’ from our Abstract (Page 1, Line 16), have added the words ‘molecular’ and ‘annotated’ to the Intro (Page 2, Lines 64 and 74), and have added the word ‘annotated’ to the Discussion (Page 10, Line 287)
to better described the major findings of our study. We believe that no further change is needed in relation to this comment.

Comment 2:

The aim / key question of the paper should be more precisely worded in the Abstract, which then allows you to better formulate a concluding sentence at the end of the Abstract. The text “The Fe homeostasis TF network is well characterised in [..] Arabidopsis and rice but remains poorly understood in hexaploid bread wheat” is rather broad and could be the basis for many research papers, or even reviews. Am I correctly assuming that IRO3 and HRZ are of interest as potential candidate genes for biofortification approaches?

We agree that this sentence within our Abstract is broad and have changed it to read ‘The Iron related transcription factor 3 (IRO3) and Haemerythrin motif-containing Really Interesting New Gene (RING) and Zinc finger protein (HRZ) proteins are major regulators of Fe homeostasis in diploid species such as
Arabidopsis (Arabidopsis thaliana) and rice (Oryza sativa L.) but remain uncharacterised in hexaploid bread wheat (Triticum aestivum L.).’ (Page 1, Lines 12-15 and associated changes to Lines 15-16). We believe this alongside our Introduction provides sufficient justification for studying these Fe homeostasis
genes in bread wheat, regardless of their relevance for biofortification approaches, and believe that no further change is needed in relation to this comment.

Comment 3:

Following on from the previous point, it is of interest to note that the core regulatory mechanism of Fe homeostasis appears to be conserved between monocots and dicots. For many decades, plants were strictly separated into Strategy I and II, because the iron uptake mechanisms were so different. Several
recent review papers by the groups of Dubos and Guerinot, see special issue of J Exp Bot that came online in the past couple of months, describe the common regulatory actors, which include IRO3 and HRZ/BTS. There may be other recent papers as well, so please use these to update refs 3 – 6 and relevant sections in the Introduction.

We agree with the Reviewer and thank them for drawing our attention to this special issue in J Exp Bot. We have added the following references to our Introduction (Page 1, Line 40) and References sections
(Page 13, Line 412-415):
7. Riaz, N.; Guerinot, M.L. All together now: regulation of the iron deficiency response. Journal of Experimental Botany. 2021, 72, 2045-2055; https://doi.org/10.1093/jxb/erab003
8. Gao, F.; Dubos, C. Transcriptional integration of plant responses to iron availability. Journal of Experimental Botany. 2021, 72, 2065-2070; https://doi.org/10.1093/jxb/eraa556
We believe that no further change is needed in relation to this comment.

Comment 4:

Line 182: It is inappropriate to claim that there is a separation between graminaceous and nongraminaceous species when the only example of the latter is banana. I would suggest you incorporate other non-graminaceous monocot species in the phylogenetic analysis, which would then provide you
with the power to make this distinction (or alternatively remove this distinction from your results).

We agree that it is inappropriate to draw conclusions based on including only one non-graminaceous monocot species (banana) in our analysis and have therefore removed this species from our phylogenetic analyses. This has resulted in changes to Figures 3 (Page 6) and 4 (Page 7), as well as any reference to banana throughout our manuscript and we believe that no further change is needed in relation to this comment.

Comment 5:
Further to point 5, it would be worth adding the BTS-like proteins in dicots to the HRZ/BTS phylogenetic tree.

No change was made in relation to this comment. We appreciate the Reviewer’s comment to add BTSlike proteins to our phylogenetic tree, but do not think these proteins are relevant to our study given the relatively low (38%) amino acid identity between BTS and BTS-like proteins in Arabidopsis, and that the
amino acid identity between BTS and the TaHRZ proteins was at least 58%.

Comment 6:
What exactly does “transcript copy number per ug of RNA” refer to in Fig 5 and 6? I have not seen this scale used for qRT-PCR previously, and it is not clear to me how this expression data incorporates normalisation by the three housekeeping genes in the methods. Can you please make this calculation
explicit in the methods, and explain the scale used in the figure captions.

We agree with the Reviewer that this y-axis scale requires explanation and have added the following sentence to the Methods section (Page 3, Lines 131-133): ‘Relative expression for each TaIRO3 and TaHRZ gene was calculated based on a standard curve of triplicate ten-fold serial dilutions (101–107) of purified
template for each primer pair and based on the amount of RNA (µg) included in the reaction.’. We have also defined ‘relative expression’ in our figure captions (Page 8, Lines 250-251 and Page 9, 257). We  believe this, combined with the next sentence in the manuscript (Page 3, Lines 134-138) describing the
use of a three gene normalisation factor (3GNF) sufficiently explains how our qRT-PCR expression values were calculated and note that similar y-axis labels have been used in the following studies:
Kobayashi, T., Nagasaka, S., Senoura, T. et al. Iron-binding haemerythrin RING ubiquitin ligases regulate plant iron responses and accumulation. Nat Commun 4, 2792 (2013). https://doi.org/10.1038/ncomms3792
Bonneau J, Baumann U, Beasley J, Li Y, Johnson AA. Identification and molecular characterization of the nicotianamine synthase gene family in bread wheat. Plant Biotechnol J. 2016 Dec;14(12):2228-2239. https://doi.org/10.1111/pbi.12577.
Broad, R.C., Bonneau, J.P., Beasley, J.T. et al. Genome-wide identification and characterization of the GDP-L-galactose phosphorylase gene family in bread wheat. BMC Plant Biol 19, 515 (2019). https://doi.org/10.1186/s12870-019-2123-1
We believe that no further change is needed in relation to this comment.

Comment 7:

You make the claim that HRZ is upregulated at 24 hours in shoot tissue after the initial iron deprivation (Figure 6), but it also looks to me that the HRZ genes are upregulated in the control environment as well, though perhaps to a lesser extent. Have you carried out comparisons across the timepoints to see whether the expression of the IRO3 and HRZ genes varies across the timecourse for both conditions? From looking at your data, it seems as if the HRZ genes in particular are upregulated 24 hours after the start of the experiment in both Fe deficient and Fe sufficient conditions, suggesting that the response may also in part be due to the transplanting/moving of the plants.

No change was made in relation to this comment. We agree with the Reviewer that the expression of TaIRO3 and TaHRZ varies over time throughout our +Fe and -Fe experiment, however this variation may be simply due to differences between plants (given our analysis was destructive) rather than treatment
duration. To account for this variation, we ran our statistical analysis (2 sample t-test) between treatments for each timepoint rather than between timepoints, as we believe this is the most valid form of statistical analysis in this situation. For example, although under control (+Fe) conditions the TaHRZ genes were
upregulated in wheat shoot tissues at Day 1 (relative to Day 0), the upregulation of these genes under -Fe conditions was nevertheless significantly higher relative to control conditions.

Comment 8:

The section looking at the KnetMiner network needs to be fleshed out more. What kinds of “associations” do you see (lines 265-267)? Looking at the network online, these seem to be based on sequence similarity so this part of the network doesn’t provide any new information beyond your phylogenetic analysis for the HRZ genes. I am also surprised that the KnetMiner network pulls out a different rice bHLH gene as most similar to the IRO3 genes. Have you looked into this bHLH062 gene any further? Is this gene equally orthologous to the wheat IRO3 genes? It would be good to know if this gene appeared in your initial BLAST identification of the IRO3 genes, and how you were able to
determine that bHLH063 (OsIRO3) is the true ortholog of the TaIRO3 genes, rather than bHLH062. It’s important to note that the majority of the data you report from the KnetMiner network is not novel (either known from your phylogeny, or is reporting knowledge from the Arabidopsis literature). The
most interesting aspect here is looking at the predicted regulation of the HRZ and IRO3 gene by the GENIE3 network. I would suggest you emphasize this aspect of the analysis much more both in the results and in the discussion.

We agree with the Reviewer that the word ‘associations’ in our Results section is ambiguous and have altered this to read ‘regulatory associations’ (Page 9, Lines 267-268). We also agree that an explanation of the bHLH062 gene in our KnetMiner network is required and provided an explanation in our Discussion
Section: ‘The KnetMiner network included a different rice bHLH protein to OsIRO3, OsBHLH062, likely due to incomplete incorporation of the rice genome database into KnetMiner. We hypothesise that the OsbHLH062 gene encodes a closely related TF to OsIRO3 in rice and warrants further investigation in future rice Fe homeostasis studies.’ (Pages 11-12, Lines, 359-362).

We agree that the predicted additional regulatory genes identified in the KnetMiner network (ABI3/VP1, etc) are the most interesting aspect and included the following phrase in our Discussion section ‘and future analysis will aim to characterize these novel TFs in bread wheat and determine whether the TaABI3/VP1 genes share similar Fe sensing roles to OsIDEF1’ to highlight our interest and future directions for this research (Page 11, Lines 357-359). We
believe that no further change is required in relation to this comment.

Comment 9:

Given your interest in the global expression of the IRO3 and HRZ genes, as well as the genome dominance they may show, I would suggest you look at the extensive expression data available on wheat-expression.com. Not only can you see expression data for the 9 genes in question, but you can visualise the homoeolog dominance using the “ternary plot” option. I’ve had a quick look at the data for your genes and it confirms your results, and would give strength to your claims about the expression profile of the IRO3 and HRZ genes.

We appreciate the Reviewer’s excellent suggestion and have added an additional Supplementary Figure 1 using the Wheat Expression Browser database. We have also added the following sentence to our Results section ‘Similar patterns of relative expression between TaIRO3, TaHRZ1, and TaHRZ2 homoeologs were observed in the Wheat Expression Browser database (Fig. S1).’ (Page 7, Lines 233-235). We have also provided a reference to the Wheat Expression Browser database in our Methods section (Page 3, Line, 138-140) and believe that no further change is required in relation to this comment.

Comment 10:

Lines 356-358: I am not sure how the GENIE3 network using RNA-Seq data is relevant to no subgroup IVc bHLH transcription factors being identified in the KnetMiner network. Also, please be aware that the GENIE3 network does not contain any iron-responsive datasets, so this would have no bearing on their inclusion in the network.

We appreciate the Reviewer’s comment and have tweaked this sentence to clarify why IVc BHLH TFs would not be detected in RNA-Seq datasets used by KnetMiner (page 12, Line 362-365) and believe no further change is needed in relation to this comment.

Comment 11:

Introduction, line 59: Wheat grows extremely well on alkaline soils and Fe uptake is never an issue, to my knowledge. The only agronomy sectors where Fe fertilization is consistently applied is for fruit trees (peach, apricot, cherry). Please rephrase this section.

We appreciate the Reviewer’s comment and have rephrased these sentences to read: ‘On alkaline soils, which comprise approximately 30% of global soils, Fe is less available for plant uptake and can lead to conditions of Fe deficiency [22]. Under Fe deficiency wheat plants exhibit leaf chlorosis and reduced yield [23].’ (Page 2, Lines 58-60). We believe this provides a distinction between alkaline soils and the effects of Fe deficiency on wheat growth and believe no further change is needed in relation to this comment.

Comment 12:

Please reference the primer supplementary table in the methods section.

We have added a reference to Table S3 (Page 3, Line 130) and believe that no further change is needed in relation to this comment.

Comment 13:

The genomes and resources used in the methods section should be cited, rather than just referenced with a URL. For example, in lines 81 and 82, include the appropriate citations for the wheat genome (Appels et al. 2018) and EnsemblPlants (Howe et al. 2019). Please include appropriate citations for the
remaining tools and genomes used in the methods section.

We have added the appropriate citations throughout our Methods section (Pages 2-3) and added the following references to our References section (Pages 13-16): [25], [36-42], [44, 50]. We believe that no further change is needed in relation to this comment.

Comment 14:

The TaIRO3-B homeologue seems to be missing the first exon. Is this really correct, or due to incomplete genome sequence?

No change was made in relation to this comment. The lack of a first exon for TaIRO3-B in Figure 1 is due to limitations of the current annotated wheat genome sequence and will likely be resolved in further iterations of wheat genome annotation.

Comment 15:

Panel C in Figures 3 and 4 is very small and difficult to read. I would suggest moving panel C from both figures into a joint supplementary figure, which allows the alignment to be larger. I would also suggest providing the given alignment as a supplementary file, which would support reconstruction of the
phylogenetic analyses.

We agree with the Reviewer and have removed panel c from Figures 3 and 4 to generate two separate figures in our Supplementary Material file (Figure S2 and Figure S3). We have also included the IRO3 and HRZ alignment files alongside our Supplementary Material file and believe that no further change is required in relation to this comment.

Comment 16:

The Discussion section is rather long given the small data set. Consider merging Results and Discussion, and restrict “Discussion” to placing the data in context, e.g. comparing the expression with exVIP data, and with recent RNA-seq data of wheat grown -Fe.

No change was made to the Manuscript in relation to this comment. In responding to the excellent comments provided by the Reviewer as outlined above (see comment 8), we have shortened our Discussion section and now believe it is appropriately sized relative to the size of our data set.

Reviewer 2 Report

Carey-Fung et al reported the wheat genes which homologous to IRO3 and HRZ which are important for regulation of Fe homeostasis in planta. As author described, wheat is important crop in the world, however the complex genome of wheat has disturbed to identify the genes which are important for plant physiology. Therefore, their attempt to identify the genes which important for Fe homeostasis by bioinformatics is meaningful.

Since wheat plants are important crop, there are many reports to identify the genes which is important for Fe homeostasis especially for Fe biofortification mainly by QTL analysis. I noticed that Sharma et al (2018) reported that IRO3 is one of a candidate gene which is important for Fe biofortification. Sharma et al (2018) reported that IRO3 is located on chromosome group 7, this is different from the present manuscript in which showed that IRO3 is located on chromosome group 2. I suggest that the present manuscript will be modified if authors discuss their data comparing with the previous QTL analysis for Fe biofortification.

*Sharma et al (2018) Precise transfers of genes for high grain iron and zinc from wheat-Aegilops substitution lines into wheat through pollen irradiation
